Association between maternal pre-delivery body mass index and offspring overweight/obesity at 1 and 2 years of age among residents of a suburb in Taiwan

Liu Hsien-Kuan
Wu Chien-Yi
Yang Yung-Ning
Wu Pei-Ling
He Zong-Rong
http://orcid.org/0000-0003-1276-8546 Yang San-Nan
Tey Shu-Leei ed104496@edah.org.tw
Department of Pediatrics, E-Da Hospital, I-Shou University , Kaohsiung , Taiwan
Derraik José
Electronic publication date: 2019 Feb 25
Publication date: 2019
Volume: 7
Electronic Location ID: e6473
Received 2018 Jun 8; Accepted 2019 Jan 18
Copyright: © 2019 Liu et al.
Copyright year: 2019
Copyright holder: Liu et al.
License: This is an open access article distributed under the terms of the Creative Commons Attribution License, which permits unrestricted use, distribution, reproduction and adaptation in any medium and for any purpose provided that it is properly attributed. For attribution, the original author(s), title, publication source (PeerJ) and either DOI or URL of the article must be cited.
License URL: https://creativecommons.org/licenses/by/4.0/

Keywords: Obesity, Maternal pre-delivery BMI, Overweight, Offspring, Body weight

Funding: The authors received no funding for this work.

==============================
Background

Overweight and obesity among children can cause metabolic syndrome in adulthood and are a significant public health issue. Some studies suggest that maternal pre-pregnancy body mass index (BMI) and excessive gestational weight gain during pregnancy are associated with overweight and obesity in offspring. However, it is difficult to collect information on accurate pre-pregnancy BMI and pregnancy weight gain for women living in areas where medical resources are scarce. Maternal pre-delivery BMI might be predictive of the risk of overweight and obesity among offspring of pregnant mothers living in suburban areas.

Methods

We retrospectively collected data on term neonates with appropriate weights for their gestational age born between April 2013 and October 2015. We excluded neonates with major congenital anomalies or diseases and incomplete data. Mothers with systemic diseases or drug abuse were also excluded. Offspring body weights and heights at 1- and 2-years-old were recorded. Maternal pre-delivery BMI was divided into following groups: <25, 25–29.9, and ≧30 kg/m2.

Results

We included 261 mother-child pairs in this study. The BMIs of the offspring differed significantly among the three maternal pre-delivery BMI groups at the age of 2 years (15.18 ± 1.04, 15.83 ± 1.28, and 16.29 ± 1.61 kg/m2, p < 0.001, respectively). After adjusting for potential cofounders possibly affecting weight using multivariate linear regression, the children’s BMIs (adjusted 95% CI: 0.71 [0.31–1.11]; p = 0.001) and BMI percentiles (adjusted 95% CI 15.80 [7.32–24.28]; p < 0.001) at the age of 2 years were significantly higher in those born to mothers with pre-delivery BMIs of 25–29.9 kg/m2 compared to mothers with pre-delivery BMIs <25 kg/m2. Maternal pre-delivery BMI ≧30 kg/m2 was significantly associated with increased BMIs (adjusted 95% CI: 1.17 [0.72–1.63]; p < 0.001) and BMI percentiles (adjusted 95% CI: 23.48 [13.87–33.09]; p < 0.001) in their children. A maternal pre-delivery BMI of 27.16 kg/m2 was the optimal cut-off for predicting offspring overweight/obesity at the age of 2 years.

Discussion

Our results indicate that the maternal pre-delivery BMI was significantly associated with offspring BMI and weight gain at the age of 2 years. A maternal pre-delivery BMI of 27.16 kg/m2 might be a useful predictor for estimating the risk of overweight or obesity in offspring at the age of 2 years.

Introduction

Overweight and obesity among children is a public health issue that can result in adult metabolic syndrome (De Onis, Blössner & Borghi, 2010; Kim, Lee & Lim, 2017). There are many factors associated with overweight/obesity in children, including early childhood lifestyle and sedentary behavior, dietary patterns, sleep patterns and durations, and parental body mass index (BMI) (Reilly et al., 2005; Dev et al., 2013). The Developmental Origins of Health and Disease (DOHaD) hypothesis, which proposed by Barker (2007), expounded the high correlation between maternal environment and their offspring. For the past few years, DOHaD hypothesis had been broadly adapted in subsequent studies concerning early life environment influences the later onset of obesity (Wadhwa et al., 2009; Kappil, Wright & Sanders, 2016). This concept has enticed an increasing number of researchers to study the associations between maternal obesity and their offspring’s obesity (Guo et al., 2015; Wrotniak et al., 2008). During the gestational period, the maternal metabolic state can affect both maternal health and fetal growth, even predisposing the offspring to the development of metabolic disorders (Catalano & Ehrenberg, 2006). As there may be a connection between maternal BMI and offspring health, this is a potentially useful means of predicting and preventing child overweight and obesity.

The Institute of Medicine (IOM) of the National Academies proposed a recommendation of appropriate gestational weight gain (GWG) based on maternal pre-pregnancy BMI (Institute of Medicine (US) and National Research Council (US) Committee to Reexamine IOM Pregnancy Weight Guidelines, 2009). Currently, studies have revealed that high pre-pregnancy BMI and excessive GWG were associated with overweight/obesity in their offspring (Li et al., 2013; Schack-Nielsen et al., 2010). In Taiwan, nearly all pregnant women deliver in a metropolitan hospital or medical provision clinic. Therefore, maternal pre-pregnancy BMI or GWG can be accessed easily during the pre-natal visits. However, for those pregnant women living in suburbs where medical resources are scarce and prenatal care was inadequate, it was difficult to collect accurate pre-pregnancy BMI and GWG (Kisuule et al., 2013; Islam et al., 2017; Debessai et al., 2016; Fobelets et al., 2015). For these women, and their future children, it became a priority to identify an alternate predictor of overweight/obesity in their offspring.

Pre-delivery BMI, which can be obtained just prior to delivery, is an easier measurement to obtain from pregnant women who lived in the suburbs. If a high pre-pregnancy BMI or excessive GWG could cause adverse pregnancy outcomes and increased offspring body weight later in life, then we considered that a high pre-delivery BMI might more or less have a similar effect. To answer this question, we designed a study to compare the association between maternal pre-delivery BMI and the BMI of the offspring at the ages of 1 and 2 years. Our goal for this study, therefore, was to find a new parameter to predict the risk of overweight and obesity in the offspring of the pregnant mothers in the suburbs, and try to find out a cut-off point of maternal pre-delivery BMI for predicting overweight and obesity in their offspring.

Materials and Methods

Study designs and subjects

E-Da Hospital is located in the Yanchao District (a suburb area in Kaohsiung), occupies 6,300 ha of land and has a population of about 29,000 people. Patients in the E-Da Hospital mostly reside in the Yanchao District and the adjacent regions. Owing to the vast territory with few residents, this district is a medical resource-limited area.

This was a retrospective cohort study. Both pregnant women and their children’s health care records from E-Da Hospital were collected. We collected 1,753 records of both the delivering mothers and their infants between April 2013 and October 2015. The children’s health records included the birth-related information (date of birth, sex, gestational weeks at time of birth, birth weight, birth height, Apgar score, among others) and the infancy health examinations during the first 24 months. Neonates who were preterm, post-term, small for GA (birth weight < 10th percentile), large for GA (birth weight > 90th percentile), with presence of major congenital anomalies, congenital disease, or incomplete data were excluded from this survey.

Pregnant women’s health records included general information (age, occupation, education, number of pregnancies, smoking habits, among others), any history of diseases, all clinical measurements (height, weight, gynecological examinations, ultrasonography, gestational diabetes mellitus (GDM) screening test, and other lab tests), complications experienced during pregnancy, and pregnancy outcomes (delivery modes, labor complications, placenta weight). The pre-delivery BMI was calculated using the weight and height recorded upon admission for delivery. BMIs were calculated by dividing the weight in kilograms by the square of the height in meters. The pre-delivery BMI was categorized into three groups: <25, 25–29.9, and ≧30 kg/m2. Mothers who were diabetic, or had pregnancy-induced hypertension, a history of illegal substance abuse, human immunodeficiency virus infected, hyper- or hypothyroidism, or incomplete medical records were all excluded.

This study included the information and clinical measurements of 261 mother-child pairs, after excluding premature infants, post-term infants, congenital anomaly, small for gestational age + large for gestational age, pregnancy-induced hypertension, GDM, GDM + pregnancy-induced hypertension, maternal substance abuse, human immunodeficiency virus infected mother, maternal hyper/hypothyrodism, and incomplete data of neonate and mother.

We obtained approval from the E-Da Hospital’s Institutional Review Board (EMRP02107N). All patient information was de-identified before analysis.

Assessment of infant body weight

All infants had a preventative health visit for vaccinations at 1 and 2 years of age according to the Taiwan vaccination programs. Body weight, body height and health status were recorded at every visit. No infants had acute illnesses at the vaccination health visits. The body weights were measured with light clothing using a digital scale, and the body heights were measured with bare feet using a recumbent length stadiometer. Weight gain was calculated as the difference between the weight at birth and weight at 1 or 2 year of age. BMIs were calculated by dividing the weight in kilograms by the square of the height in meters. The definition of children obesity is according to new growth charts for Taiwanese children (Chen & Chang, 2010). Those with missing data on body weight at 1 or 2 year of age were excluded.

Statistical analysis

Data were analyzed using SPSS statistical software version 20 (IBM Corp., Armonk, NY, USA). Proportions are presented for categorical variables and means ± standard deviations are presented for continuous variables. The Chi-square test (for categorical variables) and Kruskal–Wallis test were used for comparisons among three groups of maternal pre-delivery BMIs. We conducted a multivariate linear regression to adjust the potential variables associated with outcomes including sex, mode of delivery, and maternal pre-delivery BMI. We report 95% confidence intervals (CI) with p-values. Covariate variables were selected from among the significant predictors (p < 0.05) between the three groups according to maternal pre-delivery BMI, as well as by the use of clinical judgment. Statistical significance was defined as a p-value < 0.05.

Receiver operating characteristic (ROC) curves were drawn to determine the optimal threshold for predicting overweight and obesity in children at 2 years of age depending on maternal pre-delivery BMI. The optimal threshold value was used as the cut-off point to determine the sensitivity, specificity, and negative predictive values (NPVs) of maternal pre-delivery BMI to detect overweight and obesity at 2 years of age.

Results

This study included the information and clinical measurements of 261 mother-child pairs, after excluding premature infants (n = 314), post-term infants (n = 7), congenital anomaly (n = 9), small for gestational age + large for gestational age (n = 274), loss to follow up after discharge (n = 84), pregnancy-induced hypertension (n = 119), GDM (n = 85), GDM + pregnancy-induced hypertension (n = 61), maternal substance abuse (n = 3), human immunodeficiency virus infected mother (n = 3), maternal hyper/hypothyrodism (n = 18), and maternal incomplete data (n = 7) (Fig. 1). Among the 261 children, 137 (52.5%) were boys. The mean gestational age was 38.97 ± 0.99 weeks, and the mean birth weight was 3.18 ± 0.28 kg. The participants were stratified into three groups according to the maternal pre-delivery BMI (<25, 25–29.9, and ≧30 kg/m2). The maternal and offspring demographic characteristics are presented in Table 1. The overall overweight and obesity rates in children aged 1 and 2 years were 10.0% (26/261) and 9.2% (24/261), respectively. The rate of overweight/obesity showed no significant difference at 1 year of age in each maternal pre-delivery BMI category (5/85 (2.5%), 11/96 (11.5%), and 10/63 (15.9%), respectively; p = 0.142). However, the rate differed significantly at 2 years of age (2/79 (2.5%), 10/85 (11.8%), and 12/57 (21.1%), respectively; p = 0.003). The infants birth weights (3.08 ± 0.25, 3.24 ± 0.28, and 3.22 ± 0.27 kg, respectively; p = 0.001) and birth BMIs (12.29 ± 0.90, 12.62 ± 1.02, and 12.76 ± 1.00 kg/m2, respectively; p = 0.002) were significantly higher in children whose maternal pre-delivery BMIs were ≥25 kg/m2 than in those whose maternal pre-delivery BMIs were <25 kg/m2. Our results also revealed that 56 infants were delivered by Cesarean section (21.5%). The rate of Cesarean section differed significantly in each maternal pre-delivery BMI category (10/90 (11.1%), 27/105 (25.7%), and 19/66 (28.8%), respectively; p = 0.011). The overweight and obesity rates at 1 year showed no significant association among the three groups (p = 0.142), whereas it differed significantly at 2 years (p = 0.003).

Figure 1 Participants’ selection after excluding neonatal and maternal factors from 1,753 neonates.

Table 1 Baseline maternal and offspring demographic characteristics according to pre-delivery maternal body mass index (BMI).

	BMI	p*	
25 kg/m2 (n = 90)	25–29.9 kg/m2 (n = 105)	≧30 kg/m2 (n = 66)	
Maternal demographics	
Maternal age				0.257	
 <35 years old	74 (82.2%)	77 (73.3%)	48 (72.7%)		
 ≧35 years old	16 (17.8%)	28 (26.7%)	18 (27.3%)		
Placenta weight	653.70 ± 151.95	663.23 ± 135.42	654.67 ± 116.97	0.549	
Parity				0.402	
 Primipara	48 (53.3%)	66 (62.9%)	38 (57.6%)		
 Multipara	42 (46.7%)	39 (37.1%)	28 (42.4%)		
Offspring demographics	
Gestational age (days)	39.02 ± 0.92	39.01 ± 0.99	38.81 ± 1.07	0.461	
Birth weight (kg)	3.08 ± 0.25	3.24 ± 0.28	3.22 ± 0.27	0.001	
Birth BMI (kg/m2)	12.29 ± 0.90	12.62 ± 1.02	12.76 ± 1.00	0.002	
Sex				0.293	
 Boy	46 (51.1%)	51 (48.6%)	40 (60.6%)		
 Girl	44 (48.9%)	54 (51.4%)	26 (39.4%)		
Mode of delivery				0.011	
 Vaginal	80 (88.9%)	78 (74.3%)	47 (71.2%)		
 Cesarean section	10 (11.1%)	27 (25.7%)	19 (28.8%)		
Apgar score					
 1 min	7.97 ± 0.18	7.92 ± 0.49	7.82 ± 0.89	0.891	
 5 min	8.98 ± 0.15	8.95 ± 0.35	8.81 ± 0.58	0.824	
Data collected	
 1-year-old (days)a	380.10 ± 31.43	377.51 ± 23.08	377.93 ± 15.67	0.222	
 2-year-old (days)b	820.32 ± 48.43	816.68 ± 51.11	813.59 ± 45.99	0.319	
1-year-old body typea	
 Normal	80 (94.1%)	85 (88.5%)	53 (84.1%)	0.142	
 Overweight + Obesity	5 (5.9%)	11 (11.5%)	10 (15.9%)		
2-year-old body typeb	
 Normal	77 (97.5%)	75 (88.2%)	45 (78.9%)	0.003	
 Overweight + Obesity	2 (2.5%)	10 (11.8%)	12 (21.1%)		
Notes:

Data are presented as means ± standard deviations or as numbers (proportion).

* p-values were analyzed using the Kruskal–Wallis (for continuous variables) and Chi-square tests (for categorical variables).

a Total 244 participants were enrolled in the analysis at 1 year of age.

b Total 221 participants were enrolled in the analysis at 2 years of age.

Offspring weight index among the three maternal pre-delivery BMI groups

The anthropometric measurements of the offspring were assessed at ages 1 and 2 years. The children’s BMIs at age 1 year were significantly different among the three maternal pre-delivery BMI groups (16.37 ± 1.15, 16.67 ± 1.11, and 17.19 ± 1.61 kg/m2, respectively; p = 0.005). The children BMI percentile (45.94 ± 25.68, 52.00 ± 25.54, and 60.07 ± 25.76, respectively) and weight gain (6.21 ± 0.87, 6.32 ± 0.78, and 6.60 ± 0.99, respectively) at age 1 years showed a significant association among the three groups (p = 0.006 and p = 0.044). Similarly, the children’s BMIs at age 2 years were significantly different among the three maternal pre-delivery BMI groups (15.18 ± 1.04, 15.83 ± 1.28, and 16.29 ± 1.61 kg/m2, respectively; p < 0.001). Additionally, the children BMI percentile (35.24 ± 24.74, 49.86 ± 27.72, and 56.68 ± 29.97, respectively) and weight gain (8.80 ± 1.13, 9.12 ± 1.29, and 9.72 ± 1.57, respectively) at age 2 years showed a significant association among the three groups (p < 0.001 and p = 0.004) (Table 2).

Table 2 Offspring weight index at 1 and 2 years of age according to maternal pre-delivery body mass index (BMI).

	Maternal pre-delivery BMI	p*	
1 year olda	<25 kg/m2 (n = 85)	25–29.9 kg/m2 (n = 96)	≧30 kg/m2 (n = 63)		
BW (kg)	9.31 ± 0.91	9.55 ± 0.87	9.83 ± 1.03	0.007	
BH (cm)	75.36 ± 2.74	75.66 ± 2.58	75.60 ± 2.35	0.861	
BMI (kg/m2)	16.37 ± 1.15	16.67 ± 1.11	17.19 ± 1.61	0.005	
BMI percentile	45.94 ± 25.68	52.00 ± 25.54	60.07 ± 25.76	0.006	
Weight gainb (kg)	6.21 ± 0.87	6.32 ± 0.78	6.60 ± 0.99	0.044	
2 years oldc	<25 kg/m2 (n = 79)	25–29.9 kg/m2 (n = 85)	≧30 kg/m2 (n = 57)		
BW (kg)	12.04 ± 1.19	12.31 ± 1.39	12.93 ± 1.62	0.004	
BH (cm)	88.80 ± 3.12	88.25 ± 3.08	89.12 ± 3.16	0.461	
BMI (kg/m2)	15.18 ± 1.04	15.83 ± 1.28	16.29 ± 1.61	<0.001	
BMI percentile	35.24 ± 24.74	49.86 ± 27.72	56.68 ± 29.97	<0.001	
Weight gaind (kg)	8.90 ± 1.13	9.12 ± 1.29	9.72 ± 1.57	0.004	
Notes:

BW, body weight; BH, body height; BMI, body mass index.

* p-value was analyzed by Kruskal–Wallis test.

a Total 244 participants were enrolled in the analysis at 1 year of age.

b The weight difference between birth and 1 year of age.

c Total 221 participants were enrolled in the analysis at 2 years of age.

d The weight difference between birth and 2 years of age.

Multivariable linear regression of factors associated with offspring body weight parameters at ages 1 and 2 years of age

We adjusted for the potential confounding factors that could have affected the children’s body weights at 1 and 2 years of age (sex, mode of delivery, and maternal pre-delivery BMI). The resulting multivariate linear regression analyses are presented in Table 3. At 1 year of age, BMIs (p < 0.001), BMI percentiles (p = 0.001), and weight gain (p = 0.015) were significantly higher in children born to mothers with pre-delivery BMIs of ≧30 kg/m2 compared to those born to mothers with pre-delivery BMIs of <25 kg/m2. At 2 years of age, BMIs (p = 0.001) and BMI percentiles (p < 0.001) were significantly higher in children born to mothers with pre-delivery BMIs of 25–29.9 kg/m2 compared to those born to mothers with pre-delivery BMIs of <25 kg/m2. Similarly, maternal pre-delivery BMI ≧30 kg/m2 was also significantly associated with an increase in the children’s BMIs, BMI percentiles, and weight gain (all p < 0.001) at 2 years of age (Table 3).

Table 3 Multivariable linear regression for factors associated with offspring body weight parameters at 1 and 2 years of age.

1-year-old (n = 244)	
BMI	Crude	Adjusted*	
Variables	B (95% CI)	P	B (95% CI)	p	
Sex	0.32 [−0.01 to 0.65]	0.053	0.28 [−0.05 to 0.60]	0.092	
Mode of delivery	0.11 [−0.29 to 0.51]	0.578	−0.00 [−0.40 to 0.39]	0.987	
Maternal pre-delivery BMI†					
 25–29.9 kg/m2	0.17 [−0.39 to 0.29]	0.769	0.31 [−0.07 to 0.68]	0.109	
 ≧30 kg/m2	0.19 [0.29–1.03]	<0.001	0.80 [0.37–1.23]	<0.001	
BMI percentile	Crude	Adjusted*	
Variables	B (95% CI)	P	B (95% CI)	p	
Sex	−2.55 [−9.14 to 4.05]	0.448	−3.34 [−9.88 to 3.17]	0.312	
Mode of delivery	3.50 [−4.49 to 11.49]	0.389	1.32 [−6.72 to 9.35]	0.747	
Maternal pre-delivery BMI†					
 25–29.9 kg/m2	3.43 [−6.71 to 6.80]	0.989	5.80 [−1.81 to 13.42]	0.134	
 ≧30 kg/m2	3.76 [3.50–18.32]	0.004	14.35 [5.72–22.97]	0.001	
Weight gain	Crude	Adjusted*	
Variables	B (95% CI)	P	B (95% CI)	p	
Sex	0.36 [0.14–0.58]	0.001	0.34 [0.12–0.55]	0.003	
Mode of delivery	0.06 [−0.21 to 0.33]	0.675	0.01 [−0.26 to 0.28]	0.943	
Maternal pre-delivery BMI†					
 25–29.9 kg/m2	0.12 [−0.29 to 0.17]	0.608	0.11 [−0.14 to 0.37]	0.379	
 ≧30 kg/m2	0.13 [0.08–0.59]	0.010	0.36 [0.07–0.65]	0.015	
2-year-old (n = 221)	
BMI	Crude	Adjusted*	
Variables	B (95% CI)	P	B (95% CI)	p	
Sex	0.20 [−0.16 to 0.56]	0.283	0.14 [−0.20 to 0.49]	0.413	
Mode of delivery	−0.12 [−0.56 to 0.33]	0.606	−0.32 [−0.74 to 0.11]	0.145	
Maternal pre-delivery BMI†					
 25–29.9 kg/m2	0.19 [−0.19 to 0.55]	0.337	0.71 [0.31–1.11]	0.001	
 ≧30 kg/m2	0.21 [0.37–1.18]	<0.001	1.17 [0.72–1.63]	<0.001	
BMI percentile	Crude	Adjusted*	
Variables	B (95% CI)	P	B (95% CI)	P	
Sex	−2.21 [−9.81 to 5.39]	0.567	−3.19 [−10.45 to 4.08]	0.388	
Mode of delivery	−2.58 [−11.82 to 6.66]	0.583	−6.80 [−15.75 to 2.15]	0.136	
Maternal pre-delivery BMI†					
 25–29.9 kg/m2	5.59 [−2.12 to 13.30]	0.154	15.80 [7.32–24.28]	<0.001	
 ≧30 kg/m2	13.96 [5.42–22.49]	0.001	23.48 [13.87–33.09]	<0.001	
Weight gain	Crude	Adjusted*	
Variables	B (95% CI)	P	B (95% CI)	p	
Sex	0.27 [−0.09 to 0.62]	0.143	0.21 [−0.14 to 0.55]	0.239	
Mode of delivery	−0.42 [−0.85 to 0.02]	0.060	−0.55 [−0.97 to −0.12]	0.012	
Maternal pre-delivery BMI†					
 25–29.9 kg/m2	−0.12 [−0.49 to 0.24]	0.503	0.32 [−0.09 to 0.72]	0.124	
 ≧30 kg/m2	0.72 [0.32–1.12]	0.001	0.92 [0.46–1.38]	<0.001	
Notes:

B, unstandardized regression coefficient; CI, confidence interval; BMI, body mass index.

* Co-variables included in the linear regression analysis: sex, mode of delivery, and maternal pre-delivery body mass index

† Maternal pre-delivery BMI was grouped as follows: <25, 25–29.9, and ≧30 kg/m2; the reference category is <25 kg/m2.

Diagnostic value of maternal pre-delivery BMI for predicting their offspring’s overweight/obesity

The relative contributor of explanatory variables to offspring overweight/obesity at 2 year of age were analyzed (Table 4). Among these variables, pre-delivery BMI remained a significant risk factors that would contribute offspring overweight/obesity at 2 year of age (OR: 1.206, 95% CI [1.10–1.33]; p < 0.001). Children overweight/obesity at 1 year of age was another risk factor (OR: 2.680) but this factor was not significant (p = 0.082). In addition, the ROC curves of maternal pre-delivery BMI for predicting offspring overweight/obesity at 2 years is shown in Fig. 2. The area under the ROC curve (AUROC) was 0.750% (95% CI [0.654–0.847]); p < 0.001) for diagnosing the offspring with overweight/obesity at 2 years, according to new growth charts for Taiwanese children (Chen & Chang, 2010). The optimal cut-off point (by Youden index) for maternal pre-delivery BMI to detect their offspring’s overweight/obesity at 2 years was 27.16 kg/m2, with a NPV of 96.69%, a sensitivity of 83.3%, and a specificity of 59.4%. We also transformed the continuous variable of maternal pre-delivery BMI into categorical variable by this cutoff point. The odd ratios of maternal pre-delivery BMI >27.16 kg/m2 was 7.312% (95%CI [2.41–22.20]; p < 0.001) (Table 4).

Table 4 Risk factors associated to children overweight/obesity at 2 years of age.

Variable	OR	95% CI	p-value	
Maternal pre-delivery BMI	1.206	[1.10–1.33]	<0.001	
Maternal pre-delivery BMI (>27.16/<27.16 kg/m2)a	7.312	[2.41–22.20]	<0.001	
Children sex (male/female)	1.304	[0.55–3.08]	0.545	
Mode of delivery (Cesarean section/Vaginal delivery)	0.738	[0.24–2.28]	0.597	
Overweight/obesity at 1 year of age	2.680	[0.88–8.13]	0.082	
Notes:

OR, Odd ratio; CI, confidence interval; BMI, body mass index.

a We transformed the continuous variables of maternal pre-delivery BMI into categorical variables by the cutoff point.

Figure 2 ROC curve of maternal pre-delivery body mass index for predicting offspring obesity at 2 years of age.

Discussion

Many studies have shown that both higher maternal pre-pregnancy BMI and greater GWG are associated with increased BMIs in early childhood (Li et al., 2013; Schack-Nielsen et al., 2010). There are few studies that have evaluated the correlation between maternal pre-delivery BMI and offspring obesity. This study indicated that maternal pre-delivery BMI might also predict the risk of children developing overweight/obesity later in life. We found that the maternal pre-delivery BMI was linearly associated with the BMI of children aged 1–2 years even after adjustment for birth weight, sex, and mode of delivery. We also noticed that the risk of childhood overweight was highest among the mothers whose pre-delivery BMI was ≧30 kg/m2.

In recent years, some studies have reported that greater maternal pre-pregnancy BMIs may have an impact on the childhood body weights in their offspring (Yu et al., 2013; Xiong et al., 2016). Our results showed similar results to these studies. We found that women with higher pre-delivery BMIs were associated with higher risks for Cesarean section, higher birth weight babies, and an increased risk of offspring with overweight/obesity in the first 2 years of life. There are a couple of advantages with using maternal pre-delivery BMI. First, maternal pre-delivery BMI can be obtained easily before delivery and can be applied in all medical facilities, even in areas with limited medical resources. Second, recall errors and biases associated with pre-pregnancy BMI could be avoided by using the pre-delivery BMI.

There are a number of possible mechanisms responsible for the association between maternal pre-delivery BMI and overweight in their offspring. The DOHaD hypothesis, also called the “Barker hypothesis,” proposed by Barker (2007), Armitage, Poston & Taylor (2008) and Zheng et al. (2014), could explain this relationship. According to this hypothesis, the energy excess in the maternal diet causes the accumulation of excess adipose tissue, which might modify DNA methylation and gene expression in their offspring (Zheng et al., 2014; Morales et al., 2014). This methylation process could result in offspring adiposity (Godfrey et al., 2011). This theory indicates that high maternal pre-delivery BMI might play an important role in offspring overweight and might contribute to the overweight epidemic among infants and children.

Historically, there has been an emphasis on promoting sufficient weight gain during pregnancy in an effort to reduce low-birth weight deliveries and adverse perinatal outcomes. In 2009, the IOM published new recommendations for weight gain during pregnancy (Institute of Medicine (US) and National Research Council (US) Committee to Reexamine IOM Pregnancy Weight Guidelines, 2009). However, obesity rather than insufficient weight gain during pregnancy became a more common problem nowadays. Between 1997 and 2007, approximately 46% of pregnant women in the US gained more weight than the IOM recommended. Additional studies showed that maternal pre-pregnancy obesity and excessive GWG were associated with greater risks of future offspring obesity. Nevertheless, the recommended GWG might not be suitable for reproductive-aged women in different countries due to variations in body physique or body composition, which differed from race and ethnicity (Wagner & Heyward, 2000). Yang et al. (2015) investigated the recommended GWG for Chinese women as recommended by the IOM, and they found that the GWG suggested by the IOM might not be helpful for Chinese women. In our study, we found that maternal pre-delivery BMI 27.16 kg/m2 was a reliable cut-off value for predicting offspring overweight/obesity at 2 years of age in Chinese reproductive-aged women with an odd ratio of 7.312. High NPV of this cut-off point can offer us a guide to educate the mother to control their BMI before delivery. If the mother can control their pre-delivery BMI less than 27.16 kg/m2, there will be a 96.69% chance that their offspring will not have overweight/obesity at 2 years of age. In addition, maternal pre-delivery BMI is easier to assess than GWG for pregnant women residing in the suburbs of Taiwan.

There are several strengths in our study. The pre-delivery BMI can be measured easily and accurately despite the different scales of medical facilities and can even be easily measured in medical resource-limited hospitals. Furthermore, this measurement would not be affected by irregular or delayed prenatal care. Thus, potential misreporting and participation bias with respect to the pre-pregnant BMI can be avoided. There are also some limitations in our study. First, this was a retrospective study, so it is possible that selection bias could have influenced our results. However, this issue was minimized by stratifying the groups according to maternal pre-delivery BMIs and adjusting the possible confounding factors in this study. Second, although we created multivariable models to adjust for the potential confounders, the sample size of this study was still small and the participants were from a single hospital located in southern Taiwan. Thus, the study result may not be generalizable to other countries or other ethnicities. Third, there are still a few possible confounders that we were not able to adjust for, including childhood nutrition and physical activity. Our sample population was mainly from a suburb where the lifestyle factors are similar, so these effects are expected to be small. Last, a large number of cases were excluded due to incomplete data. However, the basal characteristics of maternal and offspring we excluded was not significantly different than those enrolling participants, suggesting that this limitation may not have introduced a significant selection bias (Supplementary Table).

Our hospital is located in a suburb area in Taiwan and, also, is the largest hospital nearby. Some infants would receive their regular health examination and vaccine administration in the local clinic instead of our hospital after birth. Therefore, anthropometric measurement and healthy status of participants could not be obtained due to loss following up. With these incomplete data, the result might not be generalized. A further large-scale study may minimize this bias.

Conclusion

In conclusion, this study result indicates that maternal pre-delivery BMI might be a new parameter to predict the risk of overweight and obesity among the offspring of pregnant mothers in a suburb of Taiwan. Furthermore, the maternal pre-delivery BMI value of 27.16 kg/m2 may suggest a useful predictor when estimating the offspring’s risk of overweight or obesity at age 2 years. These findings indicate that more attention needs to be paid to infants of mothers with higher pre-delivery BMIs, especially those ≧27.16 kg/m2.

Supplemental Information

Supplemental Information 1 The raw data for second revision.

Click here for additional data file.

Supplemental Information 2 The raw data for second revision (the exclusion data).

Click here for additional data file.

Supplemental Information 3 The comparison of basal characteristics of maternal and offspring we excluded with those enrolling participants.

Click here for additional data file.

Supplemental Information 4 Raw data for revised manuscript.

Click here for additional data file.

Supplemental Information 5 The original raw data for this study.

Click here for additional data file.

Supplemental Information 6 Baseline maternal and offspring demographic characteristics according to pre-delivery maternal body mass index (BMI) of newly added participants.

We analyze the additional 103 basal characteristics and find that there is no difference comparing to former 101 basal characteristics data. Therefore, these 103 records can be generalized into our former study population. Basal characteristics of these 103 mother-children pairs was shown in newly added supplementary table.

Click here for additional data file.

The authors are grateful to Chun-Hua Yang and Tzu-Shan Chen, who were consultants on statistical methods in this study.

Additional Information and Declarations

Competing Interests

Author Contributions

Human Ethics

Data Availability

The authors declare that they have no competing interests.

Hsien-Kuan Liu conceived and designed the experiments, performed the experiments, analyzed the data, prepared figures and/or tables, authored or reviewed drafts of the paper, approved the final draft.

Chien-Yi Wu performed the experiments, authored or reviewed drafts of the paper, approved the final draft.

Yung-Ning Yang performed the experiments, prepared figures and/or tables, authored or reviewed drafts of the paper, approved the final draft.

Pei-Ling Wu analyzed the data, contributed reagents/materials/analysis tools, authored or reviewed drafts of the paper, approved the final draft.

Zong-Rong He analyzed the data, contributed reagents/materials/analysis tools, authored or reviewed drafts of the paper, approved the final draft.

San-Nan Yang conceived and designed the experiments, contributed reagents/materials/analysis tools, authored or reviewed drafts of the paper, approved the final draft.

Shu-Leei Tey conceived and designed the experiments, performed the experiments, prepared figures and/or tables, authored or reviewed drafts of the paper, approved the final draft.

The following information was supplied relating to ethical approvals (i.e., approving body and any reference numbers):

We obtained approval from the E-Da Hospital’s Institutional Review Board (EMRP02107N). All patient information was de-identified before analysis.

The following information was supplied regarding data availability:

The raw data are available in the Supplemental Files.

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
