# Peer review of "Association between maternal pre-delivery body mass index and offspring overweight/obesity at 1 and 2 years of age among residents of a suburb in Taiwan"

_PeerJ, doi:10.7717/peerj.6473_

## Round 0.1 · original submission · Major Revisions

While the Reviewers consider your work to be of interest, both have raised a number of issues that need to be addressed.

·

Basic reporting

1) Abstract: (line 3-42), and in the reporting of the results please consider reporting the adjusted differences in the BMIs and BMI percentiles, rather than just the p-values.

2) Introduction: (line 63), this sentence needs rewording as it is not written in clear English. You cite a recent paper (2016) and suggest that the developmental origins of health and disease has been used "for the past few years". This hypothesis was established by Barker over 30 years ago. You need to make it clear whether you mean the application of this hypothesis to obesity is more recent, or are referring to the original hypothesis.
This article may be useful: Wadhwa PD, Buss C, Entringer S, Swanson JM. Developmental Origins of Health and Disease: Brief History of the Approach and Current Focus on Epigenetic Mechanisms. Seminars in reproductive medicine. 2009;27(5):358-368. doi:10.1055/s-0029-1237424.

3) Introduction (line 65), remove the word "hypothetical", as a theory is a hypothesis.

4) Introduction (line 69/71), reword this as it doesn't read quite right. Try something like "As there may be a connection between maternal BMI and offspring health, this is a potentially useful means of predicting and preventing child overweight and obesity."

5) Introduction (line 81/83), change to "For these women, and their future children, it became a priority to identify an alternate predictor of overweight/obesity in their offspring."

6) Materials and methods: Consider putting the information on participants first and measures second. Line 110/112 Consider including numbers and percentages, also please change to "Mothers who were diabetic...... ".

7) Results (line 167/169), for all other variables you report the rates, please report the percentage overweight and obese by maternal BMI grouping as you do for cesarean section delivery.

8) Discussion (line 238), when you say previously do you mean in Taiwan or do you mean historically? It is unclear.

9) Discussion (line 246), body habitus is jargon, please clarify.

Experimental design

1) It is not clear that an aim of the paper is to determine a pre-delivery BMI cut point for estimating the risk of overweight or obesity in offspring. Therefore the suggested BMI cut point of 27.15 kg/m2 is surprising. You need to set this up as an aim of the paper more clearly. Lines 90/91 of the introduction I think are suggesting you will try and find a BMI cut off value, but it is unclear.

2) This is the most important comment to be addressed. Out of 1753 records, you only include 101 records in your analysis. Many of these records are cut because you have conducted a complete case analysis. In the text you write that 1331 are excluded because they are missing information on any variable required for analysis. There is insufficient recognition of this fact in your discussion of the results, and the conclusions that you draw. Of those with incomplete data who were excluded, how many had some information, so that you could compare their characteristics with those included in the final analysis. If those who are excluded differ in important ways than those who are included, then your findings may not be generalisable. I think it is highly likely that the sample you analyse is not representative of the group you describe.

3) Expanding on the point above, please provide a short rationale for why you only wanted to look at term children with appropriate weight for GA.

Validity of the findings

The conclusion is overstated and the authors need to be more cautious with their interpretation. The small group of people you end up analysing is likely not representative of the 'healthy mothers who give birth at term in a given hospital to children of appropriate weight for gestational age' you try and study. There is a huge problem with missing data in this study that is not discussed in the strengths and limitations section of the discussion, and that is not addressed in the analysis. Please provide a discussion of the large number of records that were excluded due to missing data and the potential impact this has on the results. If these mothers and their children were more likely to be overweight for example, could it mean the BMI cut off value that you propose may change? Just how sensitive is the BMI cut off value? If possible provide some analysis to show how those included in the final analysis compare to those excluded (you may have complete information for mothers where the child information is missing for example).

Additional comments

I think this is a really novel and important piece of work. A lot of the comments I have given suggest minor changes. However, I have major concerns about the large number of records that were excluded due to missing data on all variables. The article would be strengthened substantially by discussing the potential implications of this, and if possible, by comparing those excluded because of missing information on all variables with those included.

Reviewer 2 ·

Basic reporting

1. Abstract; I suggest authors to include point estimates (beta coefficients) with 95%CI
2. Weight gain calculation should be reported in the methodology section.
3. Results; Most of statements form the first paragraph are either duplicates of the methods section or should have been taken to the methods section. For instance, line 154-155, 157-158.
4. At times authors confuse key the exposure and confounders adjusted for. For instance, line 214-217 “We found that the maternal pre-delivery BMI was linearly associated with the BMI of children aged 1-2 years even after adjustment for birth weight, sex, mode of delivery and maternal pre-delivery BMI.”

Experimental design

1. There is minimal information about the infant BMI calculations and classifications. BMI is in children is highly sex and age specific. Thus, it is not clear whether the infant BMI take into account sex and age; whether the CDC or WHO child growth chart was used. How were children categorised into overweight or obese?
2. Authors need to justify use of WHO BMI cut-offs for pregnant women. BMI calculated at the end of pregnancy does not show the level of adiposity.
3. It is unclear how confounders or covariates were chosen. For instance, birthweight may be on the causal pathway between pre-delivery BMI and the infant BMI. Adjusting for mediator variable is not appropriate analysis.

Validity of the findings

1. I agree with authors that gathering pre-delivery weight may be feasible in most settings, but it has limited implications: it is too late to prevent maternal obesity and the possible intrauterine effects.
2. The conclusion is drawn from a small highly selective sample. Only a small proportion of the original sample of children was included.

Additional comments

This study has examined the association between maternal pre-delivery body mass index and offspring overweight/obesity at 1 and 2 years of age among residents of a suburb in Taiwan. As childhood obesity is a serious public health concern, the analysis may provide some evidence for obesity prevention in children.

---

## Round 0.2 · Major Revisions

I agree with the Reviewer's comments, and strongly encourage the authors to address the issues raised.

·

Basic reporting

Abstract line 25: change "weight gain in pregnant women" to "weight gain for women"

Abstract line 35: The abstract still reads 101 mother-child pairs, instead of 204.

Discussion line 236: change "that have evaluate" to "that have evaluated"

Discussion line 295: You refer to breastfeeding as a confounder, would it not be a mediator? Overweight mothers are less likely to breastfeed and cease breastfeeding sooner than non-overweight mothers for a variety of social and physical reasons - this in turn can increase the probability that the child will have a higher BMI.
Maternal overweight-->breastfeeding-->child BMI. Care needs to be taken when deciding what is classified as a confounder and what is classified as a mediator - this is needed throughout the manuscript.

Discussion line 297: This sentence is difficult to understand. I think you want to say there was a large amount of missing data, or that a large number of cases were excluded due to incomplete data.

Experimental design

I still have two major concerns with this paper, which should be straightforward to address.

Major concern 1)
I understand that you have restricted the data to create a cohort of "healthy children". However, there are still 102 children excluded because they were lost to follow up after discharge, 64 children excluded because of incomplete medical records, and 7 children excluded because they were missing some maternal information. Some of these children would have contributed to the sample of "healthy children" if there were not missing data. You have some data on these children. It would be interesting to know how the excluded children differ in terms of the characteristics that you do have data on, compared to the children included in the sample. At the very least you need this information for the discussion to make a case as to why we should believe that the findings from your 204 children represent the group they are supposed to.

Major concern 2)
It is not clear what the purpose of the multiple regression model is and you need to state the purpose clearly in the text.

If the purpose is to assess the association between maternal BMI and child BMI/weight, then birth weight is not a confounder - it is a mediator. Mediators and confounders should be handled differently. You will underestimate the total effect of maternal BMI on child BMI if you control for variables on the causal pathway.

If the purpose is to get the best possible prediction of child BMI/weight, then none of the variables are 'confounders' they are all predictors contributing to the prediction.

I suspect the purpose is to understand the association between the exposure (maternal BMI) and the outcome (child BMI/weight). Therefore you need to identify confounders and control for these only. Anything on the causal pathway - mediators - should only be controlled for if assessing the mediating pathway or trying to estimate the 'direct' effect. If you are controlling for mediators, then delivery mode (caesarean delivery) also needs to be considered as you have it in your data set and you falg it int he discussion as an important variable.

reporting of the 27.15 cut point
Given the positive predictive value of 20% and the specificity of 58%, I believe you should include a more thorough discussion of the merits of using this as a cut point in your discussion section. The majority of children born to mothers with a BMI greater than 27.15 are not obese. But the prevalence of obesity is 5-fold higher among this group. I think there is scope here to add more useful information that just the cut point. Resources might be better targeted to towards women who have a BMI > 27.15 and other characteristics, such as a child with a large birth weight, caesarean delivery and so forth....

Validity of the findings

no comment

Additional comments

Hello,

Thank you for the effort that has gone in to making the changes to the manuscript. As you will see, I still have some comments/suggestions to improve this manuscript. My comment about the characteristics of the children excluded due to incomplete information was not sufficiently addressed in the revision. Thank you again for the opportunity to review this manuscript. I believe it is an interesting piece of research.

---

## Round 0.3 · accepted · Accept

Thank you for your efforts in addressing the reviewers' comments.

·

Basic reporting

The reporting is clear and sufficient context is provided. Figure 1 is very helpful for determining who is included/excluded and why. Overall, the manuscript is much clearer now.

Experimental design

The research question is well defined and the analyses are appropriate. The methods are described in sufficient detail for replication.

Validity of the findings

The expanded discussion on the limitations of this study has improved this manuscript. The findings, and the more cautious conclusions drawn from these findings are appropriate.

Additional comments

I have reviewed this manuscript twice already. The authors' have been very responsiveness to previous comments made and I have no further comments to make.

Reviewer 2 ·

Basic reporting

None.

Experimental design

None.

Validity of the findings

None.

Additional comments

I have no further comments.